# The Zn_1−x_Pb_x_Cr_2_Se_4_—Single Crystals Obtained by Chemical Vapour Transport—Structure and Magnetic, Electrical, and Thermal Properties

**DOI:** 10.3390/ma15155289

**Published:** 2022-07-31

**Authors:** Izabela Jendrzejewska, Tadeusz Groń, Joachim Kusz, Zbigniew Stokłosa, Ewa Pietrasik, Tomasz Goryczka, Bogdan Sawicki, Jerzy Goraus, Josef Jampilek, Henryk Duda, Beata Witkowska-Kita

**Affiliations:** 1Institute of Chemistry, University of Silesia in Katowice, 40-007 Katowice, Poland; ewa.pietrasik@us.edu.pl; 2Institute of Physics, University of Silesia in Katowice, 40-007 Katowice, Poland; tadeusz.gron@us.edu.pl (T.G.); joachim.kusz@us.edu.pl (J.K.); bogdan.sawicki@us.edu.pl (B.S.); jerzy.goraus@us.edu.pl (J.G.); henryk.duda@us.edu.pl (H.D.); 3Institute of Materials Science, University of Silesia in Katowice, 40-007 Katowice, Poland; zbigniew.stoklosa@us.edu.pl (Z.S.); tomasz.goryczka@us.edu.pl (T.G.); 4Faculty of Natural Sciences, Comenius University, 842 15 Bratislava, Slovakia; josef.jampilek@gmail.com; 5Katowice Branch, Research Network Łukasiewicz—Institute of Mechanised Construction and Rock Mining, 02-673 Warszawa, Poland; b.witkowskakita@gmail.com

**Keywords:** chalcogenides, X-ray study, magnetic and electrical properties, specific heat, thermal analysis

## Abstract

Monocrystalline chalcogenide spinels ZnCr_2_Se_4_ are antiferromagnetic and semiconductor materials. They can be used to dope or alloy with related semiconducting spinels. Therefore, their Pb-doped display is expected to have unique properties and new potential applications. This paper presents the results of dc and ac magnetic measurements, including the critical fields visible on the magnetisation isotherms, electrical conductivity, and specific heat of the ZnCr_2_S_4_:Pb single crystals. These studies showed that substituting the diamagnetic Pb ion with a large ion radius for the Zn one leads to strong short-range ferromagnetic interactions in the entire temperature range and spin fluctuations in the paramagnetic region at *H*_dc_ = 50 kOe.

## 1. Introduction

Magnetic and semiconductor spinels are of great interest because of their exotic phenomena and the fascinating ground states observed at the turn of the 20th and 21st centuries, such as [1]: behaviour of heavy fermions [2,3], complex spin order and spin dimerization [4,5,6], spin-orbital fluid [7,8], orbital glass [9], the coexistence of ferromagnetism and ferroelectricity [10,11,12], the so-called Jahn–Teller effect [13,14,15], and phenomena attributed to the competing exchange of ferromagnetism (FM) and anti-ferromagnetism (AFM) of almost equal strength, causing a strong bond frustration [16].

Many interesting properties were observed in the spinel series that fulfil Vegard’s law in the full-concentration region. In the Katowice group at the end of the 20th century, the Zn_1−x_Cu_x_Cr_2_Se_4_ spinel series was particularly intensively studied, the edge spinels of which ZnCr_2_Se_4_ and CuCr_2_Se_4_ were respectively antiferromagnet and ferromagnet, and at the same time, a semiconductor and a conductor [17,18,19]. In the early 2000s, Parker et al. [20] discovered a transition from positive to negative magnetoresistance there.

Single crystals are used in modern, innovative technology and the electronic industry. The semiconducting [21] and antiferromagnetic [22] ZnCr_2_Se_4_ spinel is a matrix for various diluted systems where the effects of the site disorder, lattice frustration, and random distribution of spin interactions [23] create novel potential applications in spin-based electronic technology. In ZnCr_2_Se_4,_ the spin order occurring at the Néel temperature *T*_N_ = 21 K [1] is an incommensurate spiral due to the narrow distances between Cr ions. Still, the larger ones cause ferromagnetic (FM) interactions confirmed by the positive Curie–Weiss temperature of θ = 90 K [21,22]. For this reason, differences in magnetic interactions between the nearest and farther neighbours in ZnCr_2_Se_4_ are related to the frustration of chemical bonds that play an essential role. The magnetic order is accompanied by structural transformation from cubic $Fd\bar{3}m$ to tetragonal *I*4_1_/*amd* symmetry with a slight contraction along the *c* axis [24,25], manifested by sharp first-order anomalies at *T*_N_ on the specific heat curve [1].

Our group’s work has been done based on the ZnCr_2_Se_4_ matrix. This matrix has been doped with both non-magnetic ions, such as, for example, Al [26,27,28], Cu [17,18,19,29,30], Ga [17,18,31,32,33], In [18,33,34], Sn [35,36], and Ta [37], and with magnetic ones, such as, for example, Dy [38,39], Gd [40], Ho [41], Mn [42,43], Nd [44], and Ni [45]. All the above-mentioned doped spinels have two features in common. They are semiconductors and antiferromagnets, just like a matrix. An interesting feature is the appearance of the re-entrant-type spin-glass when non-magnetic Al [27] and Sn [36] ions are located in the octahedral sites of the spinel structure. It was also observed that Sn ions occupying octahedral sites reduce the *n*-type electrical conductivity by magnitude compared to samples in which they occupy tetrahedral ones [35]. This interesting phenomenon has been explained based on the quantum band model, which predicts a lowering of the Fermi level in the lowest Cr^3+^ Mott–Hubbard sub-band with a narrowed 3*d*^3^ *t*_2g_ band, which gives an increase in the energy gap when Sn^3+^ ions (spin defects) are replaced at octahedral sites instead of Cr^3+^ ones [35]. Interestingly, geometric spin-glass was observed when substituting Dy ions, a paramagnetic rare-earth ion in both tetrahedral and octahedral positions [39].

This work continues our research that involved introducing the p-block element into the crystal lattice ZnCr_2_Se_4_. Pb belongs to group 14 of the periodic table. Sn also belongs to this group. ZnCr_2_Se_4_ single crystals doped with tin showed the phenomenon of spin-glass [36]. It was assumed that the presence of Pb ions in the ZnCr_2_Se_4_ crystal lattice would affect the physicochemical properties of the parent compound. A single crystal growth model was developed using a computer programme. The structural, magnetic, electrical, and thermal properties were determined using various methods (XRD—X-ray Diffraction, SEM—Scanning Electron Microscopy, DSC—Differential Scanning Calorimetry, TG—Thermogravimetry, SQUID—Superconducting Quantum Interference Device, QD-PPMS—Quantum Design Physical Property Measurement System). This work presents the structural, magnetic, electrical, thermal, and specific heat properties of five monocrystalline samples doped with diamagnetic Pb^2+^ ions with a large ion radius instead of Zn^2+^ ones. The novelty of this work is the study of the ac magnetic susceptibility measured in an external dc magnetic field up to 50 kOe. According to our knowledge, such crystals have not been described in the literature.

## 2. Materials and Methods

High-purity elements, Zn, Pb, Cr, Se (5N, Sigma Aldrich, Sofia, Bulgaria), and anhydrous CrCl_3_ (Sigma Aldrich), were applied to synthesise ZnCr_2_Se_4_ single crystals doped with Pb.

### 2.1. Computation of Thermodynamic Parameters for the Chemical Transport Reactions

The basis for the growth of single crystals by gaseous chemical transport is reversible transport reactions. The criterion determining the reaction’s transport capacity is the order of magnitude of the equilibrium constant *K_p_*. The *K_p_* value should be close to 1 (*logK_p_* ≈ 0). The following dependence is used to assess the transport capacity of the reaction:(1)$$\Delta G_{T}^{0}=\Delta H_{298K}^{0}-T\Delta S_{298K}^{0}$$
where $\Delta H_{298K\;}^{0}$ and $\Delta S_{298K}^{0}$ are the corresponding changes of the reaction’s standard and enthalpy and entropy. Free energy variations can be used to assess the transportability of the reaction $\Delta G^{0}$ instead of the equilibrium constant *K_p_* because these quantities are related to the formula:(2)$$\Delta G^{0}=-2.303RTlogK_{p}=\Delta H^{0}-T\Delta S^{0},$$
where: *R*—gas constant; *T*—absolute temperature; *K_p_*—equilibrium constant; $\Delta G^{0}$, $\Delta H^{0}$, $\Delta S^{0}$—changes in the reaction’s free energy, enthalpy, and entropy. Accompanying reactions related to the solvent or reaction products’ dissociation may occur during the transport process. Therefore, all independent reactions occurring in the tested system should be considered. The selection of an appropriate transport substance involves finding such transport reactions for which the values of *K_p_* or $\Delta G^{0}$, in the selected temperature range, do not deviate too much from the equilibrium values. We are looking for a substance that will produce sizeable partial pressure differences Δ*p_i_*, with the slightest possible temperature difference.

In the process of selenium chromite single crystal growth, anhydrous CrCl_3_ is used, which dissociates at a temperature above 773 K according to the following reactions [46,47]:2CrCl_3(g)_ = 2CrCl_2(g)_ + Cl_2(g)_,
CrCl_2(g)_ + Cl_2(g)_ = CrCl_4(g)_.

In a closed system, in a gas solution, apart from CrCl_3_ and Cl_2_, there is also CrCl_4_. Therefore, the dissociation products should be taken into account when creating a set of all independent transport reactions that occur simultaneously in the ZnSe–PbSe–CrCl_3_ system:2ZnSe + 4CrCl_3(g)_ = 2ZnCl_2(g)_ + 4CrCl_2(g)_ + Se_2(g)_2ZnSe + 4CrCl_3(g)_ = Zn_2_Cl_4(g)_ + 4CrCl_2(g)_ + Se_2(g)_2PbSe + 2CrCl_3(g)_ = 2PbCl(g) + 2CrCl_2(g)_ + Se_2(g)_2PbSe + 4CrCl_3(g)_ = 2PbCl_2(g)_ + 4CrCl_2(g)_ + Se_2(g)_2ZnSe + 2CrCl_4(g)_ = 2ZnCl_2(g)_ + 2CrCl_2(g)_ + Se_2(g)_2ZnSe + 2CrCl_4(g)_ = Zn_2_Cl_4(g)_ + 2CrCl_2(g)_ + Se_2(g)_2PbSe + CrCl_4(g)_ = 2PbCl_(g)_ + CrCl_2(g)_ + Se_2(g)_2PbSe + 2CrCl_4(g)_ = 2PbCl_2(g)_ + 2CrCl_2(g)_ + Se_2(g)_2ZnSe + 2Cl_2(g)_ = 2ZnCl_2(g)_ + Se_2(g)_2ZnSe + 2Cl_2(g)_ = Zn_2_Cl_4(g)_ + Se_2(g)_2PbSe + Cl_2(g)_ = 2PbCl_(g)_ + Se_2(g)_2PbSe + 2Cl_2(g)_ = 2PbCl_2(g)_ + Se_2(g)_

These hypothetical reactions were used to calculate the temperature dependence of the thermodynamic parameters (Δ*H*^0^, Δ*G*^0^, Δ*S*^0^, and *logK_p_*). The calculations were performed using the H.S.C. Chemistry 6 computer programme. Based on these calculations, the reaction conditions (dissolution zone temperatures and crystallisation and their difference (Table 1)) make it possible to obtain good-quality crystals.

### 2.2. Synthesis of ZnSe and PbSe

The polycrystalline ZnSe and PbSe were obtained by the high-temperature sintering of powders (1073 K) using a solid-state reaction in a high vacuum (10^−5^ mbarr).

### 2.3. Crystal Structure and Chemical Composition

The X-ray diffraction measurement was carried out at 293(1) K. Five samples with different nominal Pb content were selected under a stereoscopic Zeiss optical microscope. A small crystal (diameter of about 0.1 mm) was mounted on a glass capillary. The data were collected using a SuperNova X-ray diffractometer with a microfocus X-ray tube, optimised multi-layer optics for Mo-Kα (λ = 0.71073 Å) radiation, and an Atlas CCD detector. Accurate cell parameters were determined and refined with CrysAlisPro software (Version 1.171.37.35, Agilent Technologies, Wrocław, Poland, 2014). Moreover, the CrysAlisPro program was used to integrate the collected data. The spinel structure (space group no. 227) was refined using the SHELXL-2013 program [48]. All atoms were refined with anisotropic displacement parameters. The chemical composition of the single crystals was studied using a Scanning Electron Microscope JEM 6480 equipped with an energy-dispersive X-ray spectrometer (SEM/EDS). The solid elements (Zn, Pb, Cr, Se) were used as standards to determine the mass % content of the elements in the sample. Measurements were carried out at 20 locations of the single crystal to determine the average chemical composition. Each of the measuring areas was approximately 50 × 30 μm. Then, the average chemical composition was calculated, which is shown in Table 1. The error bar represents the standard deviation. Relatively low values of the standard deviation indicate good homogeneity of the chemical composition.

### 2.4. Magnetic and Electrical Measurements

Dynamic ac magnetic susceptibility, χ_ac_, was measured in the temperature range 2–300 K and at an internal oscillating magnetic field *H*_ac_ = 1 Oe, with an internal frequency *f* = 120 Hz taken at external static dc magnetic fields *H*_dc_ = 0, 10, 20, 30, 40, and 50 kOe. Magnetisation isotherms, *M*(H), were measured at 2, 10, 20, 40, 60, and 300 K and in the static magnetic field up to 70 kOe. A Quantum Design MPMS-XL-7AC SQUID magnetometer (Quantum Design, San Diego, CA, USA) was used. The Néel (*T*_N_) and *T*m temperatures and the critical fields were defined as the extremes corresponding to the derivative of dχ_ac_/d*T* vs. *T* and d*M*/d*H* vs. *H*. The effective magnetic moment was calculated using the equation [37,42,49]: $\mu_{eff}=2.828\sqrt{C}$, where *C* is the Curie constant. The magnetic super-exchange integrals for the first two coordination spheres, *J*_1_ and *J*_2,_ were calculated using the Holland and Brown equations [50]: $J_{1}=\left(-9T_{N}+\theta \right)/60$ and $J_{2}=\left(3T_{N}+\theta \right)/120$. The electrical conductivity, σ(T), was measured by the dc method using a KEITHLEY 6517B Electrometer/High Resistance Meter (Keithley Instruments, L.L.C., Solon, OH, USA) in the temperature range of 77–400 K. The activation energy, *E*_a_, was determined below room temperature and in the temperature range of 300–400 K from the formula $\sigma =\sigma_{0}\exp(-\frac{E_{a}}{kT})$, where *k* is the Boltzmann constant and σ_0_ is the reference conductivity. The details of methods used to determine the electrical properties, thermal analysis, and specific heat are described in [37,39,41,44].

## 3. Results and Discussion

### 3.1. Growth of Single Crystals and Chemical Composition

Thermodynamic calculations indicate that, for the ZnSe–PbSe–CrCl_3_ system, the simultaneous transport of ZnSe and PbSe will be realised using gaseous CrCl_3_ and CrCl_4_ (*logK_p_* and Δ*G*^0^ values are close to zero in the selected temperature range: 1000–1400 K). It is shown in Figure 1 and Figure 2 that, for chemical reactions with CrCl_3_ and CrCl_4,_ the enthalpy values are positive *(*Δ*H*^0^ > 0), which indicates that the transport takes place from a higher temperature to a lower temperature in the direction of the formation of products.

On the other hand, the enthalpy values for the transport of reactions with chlorine (Cl_2_) are negative *(*Δ*H*^0^ < 0), which indicates that the reaction equilibrium is shifted towards the reactants. Based on these calculations, it has been confirmed that in the ZnSe–PbSe–CrCl_3_ system, ZnSe and PbSe are mainly transported by gaseous CrCl_3_ and CrCl_4_. Values of *logK_a_* are close to zero in the selected temperature range: 1000–1400 K (Figure 3).

Synthesis of single crystals of Zn_1–x_Pb_x_Cr_2_Se_4_ was carried out according to the reaction:4(1 − x)ZnSe + 4xPbSe + 2CrCl_3_ = Zn_1−x_Pb_x_Cr_x_Se_4_ + 3(1 − x)ZnCl_2_ + 3xPbCl_2_
for x = 0.1, 0.2, 0.3.

Thermodynamic calculations determined the reaction conditions, such as dissolution and crystallisation temperatures, and their difference. The crystallisation zone temperature was between 1083 and 1153 K. The dissolution zone was between 1133 and 1233 K. Their difference was 50–70 K. Based on our experience with the growth of doped ZnCr_2_Se_4_ single crystals, we can state that a minor temperature difference (ΔT) is more favourable for crystal growth. Stoichiometric mixtures of the ZnSe, PbSe, and CrCl_3_ were introduced to quartz ampoules (length—200 mm, inner diameter—20 mm) evacuated to 10^−5^ mbarr. The sealed ampoules were placed in a two-zone tubular furnace and heated for 450 h. After heating, the ampoules were cooled at about 70 degrees per hour. Such a preparation process for the growth of single crystals allowed for obtaining single crystals of good quality (Figure 4). Five single crystals with various Pb content were chosen for X-ray diffraction measurements. The reaction conditions and results of SEM analysis, together with determined chemical formulae, are shown in Table 1.

### 3.2. Structural Study

Based on the structure of ZnCr_2_Se_4_, in which the Zn^2+^ ions occupy the tetrahedral position 8a: 1/8, 1/8, 1/8 (A-site), and the Cr^3+^ ions occupy the position 16d: ½, ½, ½ (B-site), according to the spinel structure, we considered two models of cation distribution. In the first model, the presence of Pb^2+^ ions in A-sites was assumed with coupled site occupancy factors (SOF) and constrained atomic displacements. The second model assumed that the Pb^2+^ ions would substitute the Cr^3+^ ones. Because of the strong correlation, the SOFs for Cr and Pb were refined separately in alternative calculations. The assumed oxidation state of Pb and the correlations of the refined parameters caused the rejection of the second model.

Based on this refinement, the first model allowed us to obtain the acceptable atomic displacement parameters and SOFs. We could describe the general chemical formula for obtained single crystals as presented in Table 2 and Table 3. The observed slight increase in thermal shift at the A-site points out a static disturbance in the location of the Zn^2+^/Pb^2+^ pseudo-ion, which is usually caused by the difference in ionic charges and in ionic and covalent radii (r_i_ (Zn^2+^) = 0.60 Å, r_i_ (Cr^3+^) = 0.62 Å, r_i_ (Pb^2+^) = 0.98 Å and R_c_ (Zn^2+^) = 0.74 Å, R_c_ (Cr^3+^) = 0.76 Å, R_c_ (Pb^2+^) = 1.12 Å, respectively) [51].

The structural study confirmed the chemical composition obtained from SEM. The general formula of obtained single crystals can be described as Zn_1−x_Pb_x_Cr_2_Se_4_. The Pb ions substituted the Zn ones. The single crystals crystallised in a cubic system with space group *Fd-*3*m* (No. 227, Z = 8). The lattice parameters of the obtained single crystals are slightly larger than pure ZnCr_2_Se_4_ (a = 10.489 Å) and enhanced with the increase in lead content (Table 2). The increase in lattice parameters is connected with the difference in ionic radii between Zn^2+^ and Pb^2+^ (Figure 5).

The positional parameter of Se (*u*) is a measure of the anion sublattice distortion from the cubic-close packing. The value of *u* is the same for all obtained single crystals. Its value increases slightly compared to the ideal value of *u* = 0.250 (Table 3). The admixture of lead does not significantly affect the *u*-value. The same is observed in the metal–metal and metal–selenium distances (Table 4), where the differences are insignificant [22]. The structure refinement parameters, the selected bond distances, and angles are shown in Table 3 and Table 4.

### 3.3. Electrical and Magnetic Properties

The results of the electrical measurements are shown in Figure 6 and Table 5, and the magnetic ones are in Figure 7, Figure 8 and Figure 9 and Table 5 and Table 6. Electrical conductivity measurements (Figure 6) showed semiconducting properties in the temperature range of 77–400 K for all single crystals under study. Two regions of Arrhenius type electrical conductivity are visible: the extrinsic one with the activation energy of *E*_a_ ~0.08 eV below room temperature, independent of the Pb content in the sample, and the intrinsic one with *E*_a_ increasing in the range of 0.138–0.315 eV above room temperature with increasing Pb content (Table 5).

The increase in activation energy with the increase in lead content may be caused by the appearance of cationic vacancies (acting as double acceptors) because the radius of Pb^2+^ ions is much larger than that of Zn^2+^ ones [51]. It may result in acceptor vacancy levels in the energy gap. The activation energies mentioned above correlate well with *E*_a_ for the matrix [31] and with its admixtures, such as tin [35], tantalum [37], and dysprosium [22].

The temperature dependencies of ac magnetic susceptibility χ_ac_, recorded in an internal oscillating magnetic field *H*_ac_ = 1 Oe with an internal frequency *f* = 120 Hz and with a zero external static magnetic field (Figure 7a–e), show AFM order with the Néel temperature *T*_N_ = 22 K and the positive Curie–Weiss temperature (θ) slightly changing from 74 K to 84 K as the Pb content increases (Table 5).

The θ values are much lower than the literature data published in [21,22] and slightly lower than in [1], where θ is equal to 115 K and 90 K, respectively. Thus, only the *T_N_* temperature values are close to this temperature’s values for the ZnCr_2_Se_4_ matrix [1,21,22]. It means that the long-range AFM interactions are slightly more substantial. In contrast, the short-range FM ones are somewhat weaker in the sample containing Pb, which is confirmed by both the long bond lengths (Table 4) and the values of the *J*_1_ and *J*_2_ super-exchange integrals for the first two coordination spheres (Table 5). A significant difference was observed between the effective magnetic moment (μ_eff_) and the effective number of Bohr magnetons, *p*_eff_ = 5.477, i.e., the spin contribution to the magnetic moment for Cr^3+^ ions per molecule 3*d*^3^ configuration. It may suggest that the orbital magnetic contribution has not been quenched at low amplitude, *H*_ac_ = 1 Oe, of the applied oscillating magnetic field.

The magnetisation measurements (Figure 8a–e) show that the magnetic saturation of the studied samples is close to the value of 6 μ_B_/f.u. for the matrix; the first critical field is kept constant at *H*_c1_ = 12 kOe, and the second critical field is around *H*_c2_ = 57 kOe (Table 5). The hysteresis loops have zero-field coercivity and zero remanences (Figure 8f).

The temperature dependencies of ac magnetic susceptibility χ_ac_, recorded in an internal oscillating magnetic field *H*_ac_ = 1 Oe with an internal frequency *f* = 120 Hz and taken at external static magnetic fields *H*_dc_ = 0, 10, 20, 30, 40, and 50 kOe, are shown in Figure 9a–e. With the increase in *H*_dc_, we observed a shift of *T*_N_ towards lower temperatures and of θ towards higher ones (Table 6).

A strong magnetic field weakens the AFM order and strengthens the FM one. Moreover, the Curie constant, *C*, and the effective moment μ_eff_, are close to the values typical for a chromium ion per molecule. For *H*_dc_ = 50 kOe, the *J*_1_ super-exchange integral for the first coordination sphere changes the sign from negative to positive, while the *J*_2_ integral remains positive (Table 6). It means that the short-range FM interaction extends over the entire temperature range. The *χ*_ac_(*T*) curves above *T*_N_ in the paramagnetic region show characteristic wide maxima at *T*_m_ = 32–36 and 44–48 K in the fields *H*_dc_ = 40 and 50 kOe, respectively. These broad maxima of ac magnetic susceptibility may be due to the spin fluctuations that occur due to the amplification of short-range FM interactions by a static magnetic field, which is opposed by the thermal energy *kT*. Similar maxima were found in the ZnCr_2_Se_4_ matrix doped with Al [28], Ce, Ga, and In [33].

### 3.4. Specific Heat Studies

In Figure 10, the points represent the specific heat measured for samples with the formulae: Zn_0.92_Pb_0.07_Cr_2_Se_4_, Zn_0.92_Pb_0.09_Cr_2_Se_4_, Zn_0.89_Pb_0.11_Cr_2_Se_4_, and Zn_0.88_Pb_0.12_Cr_2_Se_4_.

We see that the Debye model fits the experimental data well for all the samples, with the Debye temperatures obtained from the fits 163–177 K, with no apparent dependence of Debye temperature on the Pb concentration. The number of atoms obtained from the fit lies within 7.2–9.5 for the investigated samples, without clear dependence on the Pb concentration. It can be explained by the fact that the change in sample composition is minute, and the changes in Zn concentration accompany the changes in Pb concentration. The upper inset of Figure 10 shows the specific heat in a narrow region around the magnetic ordering temperature. The sample Zn_0.92_Pb_0.09_Cr_2_Se_4_ exhibits quite a sharp peak in the vicinity of magnetic transition, whereas the remaining compositions show a broader two-peak-like structure. This is most likely caused by the disorder induced by the Pb substitution and the lack of full site occupancy of the Zn site. The bottom inset of Figure 10 shows the shift of the magnetic transition with the increase in magnetic field for the Zn_0.92_Pb_0.07_Cr_2_Se_4_ sample, which is a clear manifestation of antiferromagnetic order.

We also measured the resistivity for the Zn_0.88_Pb_0.12_Cr_2_Se_4_ sample (the sample with the highest Pb concentration) with the resistivity option in the PPMS instrument (Figure 11). Below 120 K, the resistivity exceeded the range of the instrument. However, in the 120–300 K range, the resistivity can be well-fitted with an activation law ρ = ρ_0_exp(Δ/kT) (ρ_0_—resistivity at 273.15 K), where for an indirect bandgap, we obtained Δ ~0.88 eV. The inset shows the resistivity measured vs. magnetic field at room temperature.

The magnetoresistivity is positive and almost linear with the magnetic field. It is also relatively small, below 2% for 9 T.

### 3.5. Thermal Analysis

The thermal analysis examined two crystals of Zn_1−x_Pb_x_Cr_2_Se_4_ containing minimum (0.06) and maximum (0.12) lead amounts. The DSC and TG curves are depicted in Figure 12.

The shape of the DSC curves is similar to pure ZnCr_2_Se_4_. For the sample Zn_0.94_Pb_0.06_Cr_2_Se_4_, two small endothermic peaks are observed at 711 °C and 733 °C. The observed peaks are in a similar position close to pure ZnCr_2_Se_4_ (755 °C) [44]. These peaks indicate that the melting process of the sample starts at about 711 °C. The peak confirms it on the DTG curve. However, the sample Zn_0.88_Pb_0.12_Cr_2_Se_4_ lacks typical endothermic peaks. On the TG curves for both samples, a few percent increase in weight is visible in the first step. This phenomenon can indicate that the investigated sample absorbs an inert gas during the heating or solid-state–gas reaction. The mass loss is observed with increasing temperature on the DTG curve. It is an endothermic reaction, as indicated by the downward deflection of the DSC curve. The partial destruction of investigated samples takes place at about 1300 °C. It suggests that the samples doped with Pb are thermally more stable than pure ZnCr_2_Se_4_.

## 4. Conclusions

In conclusion, we have demonstrated the family of Zn_1−x_Pb_x_Cr_2_Se_4_ (x = 0.06, 0.07, 0.09, 0.11, 0.12) spinel single crystals showing semiconducting and AFM behaviour. These single crystals were successfully obtained using chemical vapour transport. The technology of crystal growth was based on thermodynamic calculations. XRD analysis showed that these crystals possess a cubic structure (SG: *Fd*3*m*). Thermal analysis showed that the single crystals under study are thermally stable up to 1300 °C.

With the increase in the external magnetic field, a shift of *T*_N_ and the specific heat peak towards lower temperatures and θ towards higher ones was observed, as well as a strong weakening of long-range AFM interactions visible in the reduction of the super-exchange integral for the first coordination sphere and the appearance of spin fluctuations in the paramagnetic region, visible in magnetic fields of 40 and 50 kOe. Below *T*_N,_ the magnetic field dependence of magnetisation, *M*(H), showed two peculiarities at critical fields *H*_c1_ = 12 and *H*_c2_ = 57 kOe and a change of the sign of the *J*_1_-integral from negative to positive at *H*_dc_ = 50 kOe, which causes the short-range FM interaction to extend over the entire temperature range. Finally, we can conclude that doped chalcogenide spinels are suitable materials for application in many technological areas, such as magnetic materials for capacitors, transformers, or components in electronic products.

## Figures and Tables

**Figure 1 materials-15-05289-f001:**
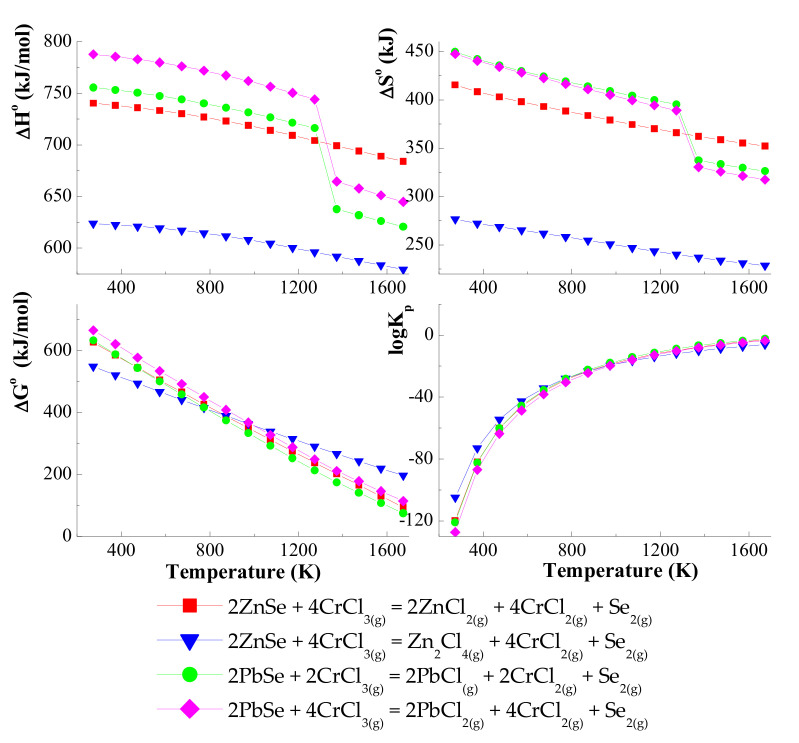
The dependence of the thermodynamic parameters vs. temperature *T* for the ZnSe and PbSe with CrCl_3_ as a transport agent.

**Figure 2 materials-15-05289-f002:**
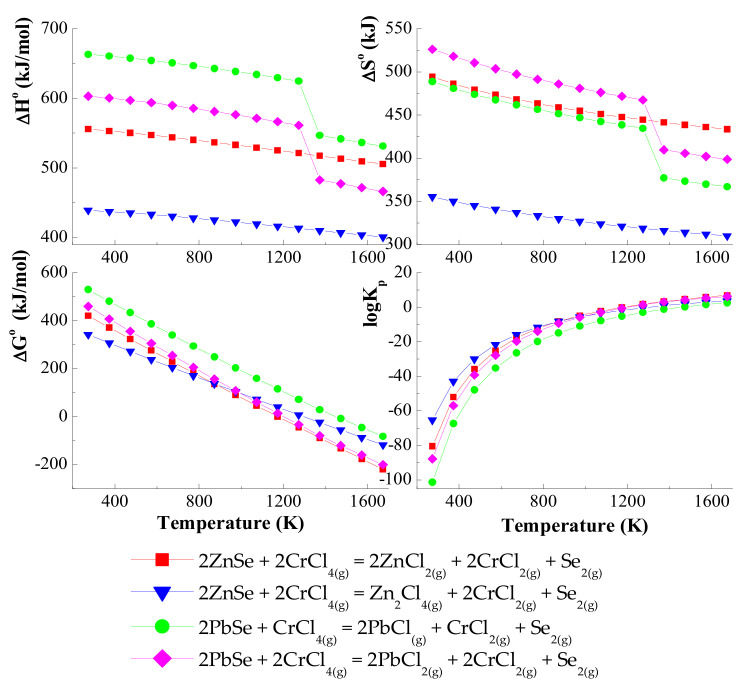
The dependence of the thermodynamic parameters vs. temperature *T* for the ZnSe and PbSe with CrCl_4_ as a transport agent.

**Figure 3 materials-15-05289-f003:**
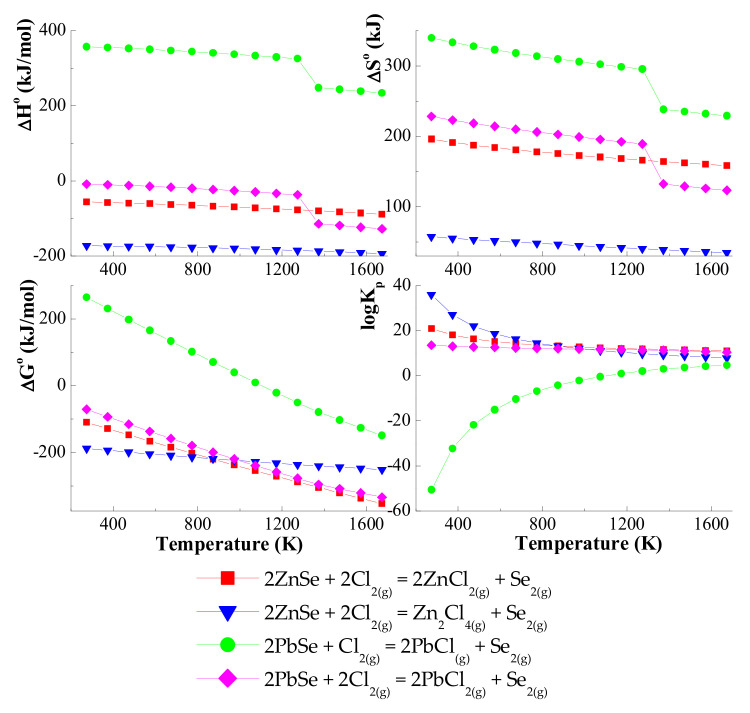
The dependence of the thermodynamic parameters vs. temperature *T* for the ZnSe and PbSe with Cl_2_ as a transport agent.

**Figure 4 materials-15-05289-f004:**
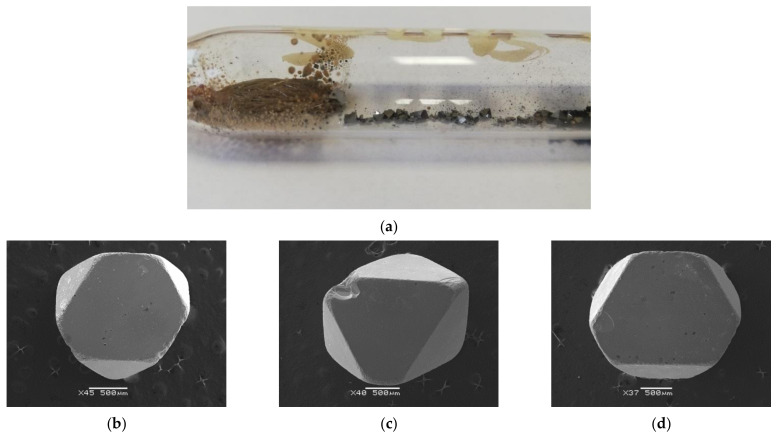
(**a**) A part of quartz ampoule with obtained ZnCr_2_Se_4_:Pb single crystals. Examples of single crystals obtained in the ZnCr_2_Se_4_:Pb system: (**b**) Zn_0.94_Pb_0.06_Cr_2_Se_4_, (**c**) Zn_0.92_Pb_0.09_Cr_2_Se_4_, (**d**) Zn_0.88_Pb_0.12_Cr_2_Se_4._

**Figure 5 materials-15-05289-f005:**
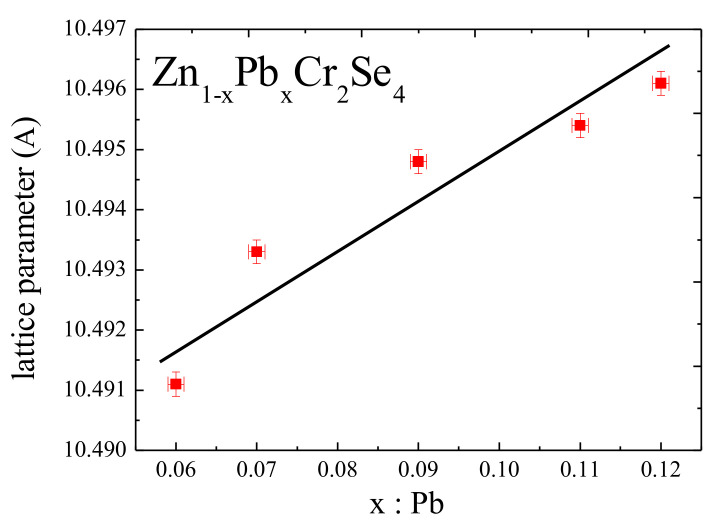
Dependence of the lattice parameters of Zn_1-x_Pb_x_Cr_2_Se_4_ single crystals on the Pb amount.

**Figure 6 materials-15-05289-f006:**
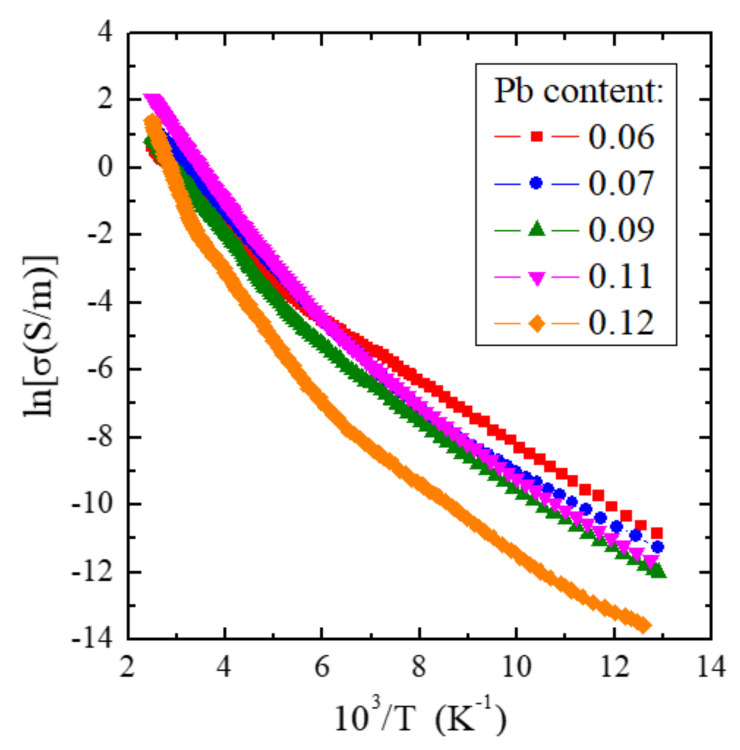
Electrical conductivity (lnσ) vs. reciprocal temperature (10^3^/T) for ZnCr_2_Se_4_ single crystals doped with Pb^2+^ ions having contents of 0.06, 0.07, 0.09, 0.11, and 0.12.

**Figure 7 materials-15-05289-f007:**
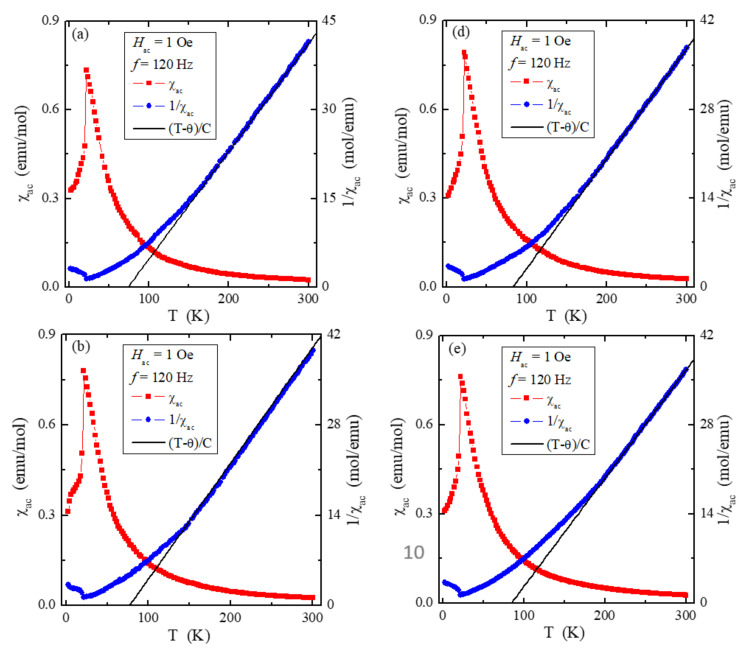
Ac magnetic susceptibility χ_ac_ and its reciprocal 1/χ_ac_ vs. temperature *T* recorded at internal oscillating magnetic field *H*_ac_ = 1 Oe with internal frequency *f* = 120 Hz for the ZnCr_2_Se_4_ single crystals doped with Pb^2+^ ions having contents of 0.06 (**a**), 0.07 (**b**), 0.09 (**c**), 0.11 (**d**), and 0.12 (**e**). The solid (black) line, *(T–*θ*)/C*, indicates a Curie–Weiss behaviour.

**Figure 8 materials-15-05289-f008:**
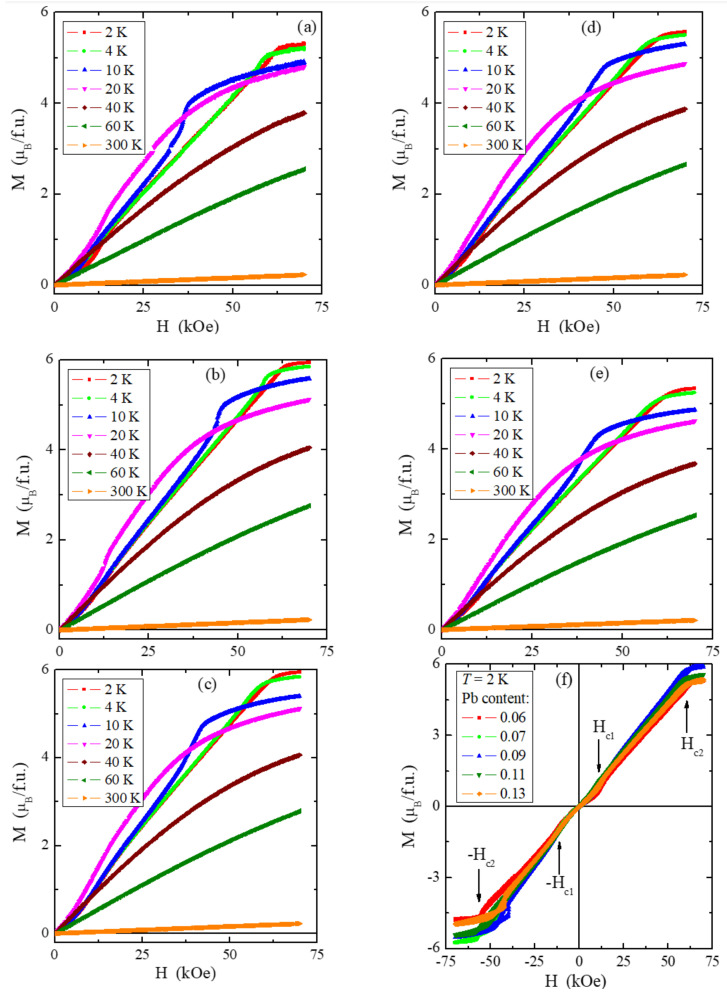
Magnetization *M* vs. magnetic field *H* at 2, 4, 10, 20, 40, 60, and 300 K for the ZnCr_2_Se_4_ single crystals doped with Pb^2+^ ions having contents of 0.06 (**a**), 0.07 (**b**), 0.09 (**c**), 0.11 (**d**), and 0.12 (**e**). The *M*(H) isotherms recorded at 2 K with arrows denoting the first (*H*_c1_) and second (*H*_c2_) critical magnetic fields are shown in figure (**f**).

**Figure 9 materials-15-05289-f009:**
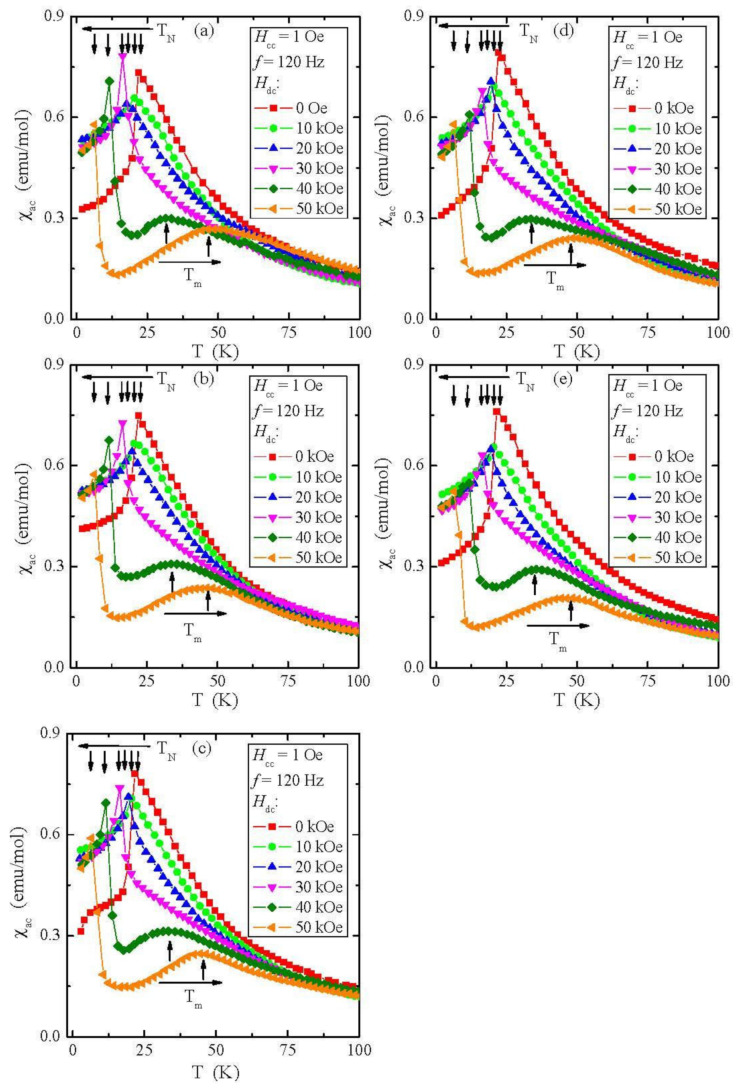
Ac magnetic susceptibility χ_ac_ vs. temperature *T* recorded at internal oscillating magnetic field *H*_ac_ = 1 Oe with internal frequency *f* = 120 Hz for the ZnCr_2_Se_4_ single crystals doped with Pb^2+^ ions having contents of 0.06 (**a**), 0.07 (**b**), 0.09 (**c**), 0.11 (**d**), and 0.12 (**e**) taken at external magnetic fields *H*_dc_ = 0, 10, 20, 30, 40, and 50 kOe. Vertical arrows indicate the Néel *T*_N_ and *T*_m_ temperatures, and horizontal ones indicate the shift of *T*_N_ and *T*_m_ with increasing magnetic field *H*_dc_.

**Figure 10 materials-15-05289-f010:**
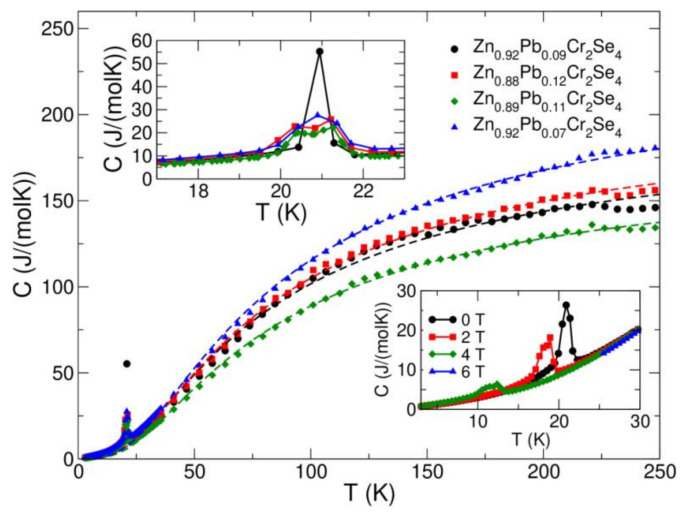
Specific heat, *C*, as a function of temperature *T*, measured for the Zn_1−x_Pb_x_Cr_2_Se_4_ single crystals (x = 0.07, 0.09, 0.11, and 0.12) at zero magnetic field (main figure). The dashed lines show the Debye model fit experimental data for *T* > 40 K. The upper inset shows how the magnetic peak is affected by Pb substitution. The lower inset shows how the magnetic transition temperature is affected by the magnetic field for the Zn_0.92_Pb_0.07_Cr_2_Se_4_ sample. Note that the temperature scales on the lower and upper inset are not the same. Hence, the peaks in the upper inset appear much broader than in the lower inset or the main figure.

**Figure 11 materials-15-05289-f011:**
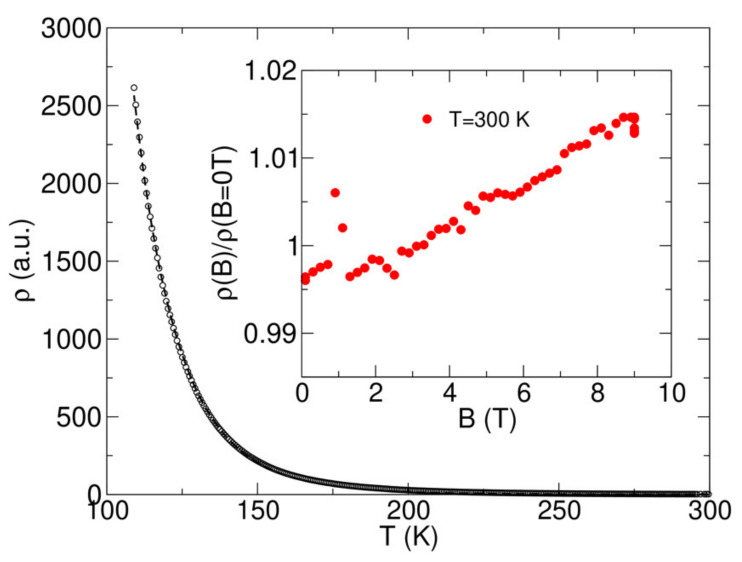
The resistivity of the Zn_0.88_Pb_0.12_Cr_2_Se_4_ single crystal measured vs. temperature (central Figure) and vs. field at T = 300 K (inset). The Zn_0.88_Pb_0.12_Cr_2_Se_4_ sample was tiny. Hence, we do not provide the absolute value of the resistivity.

**Figure 12 materials-15-05289-f012:**
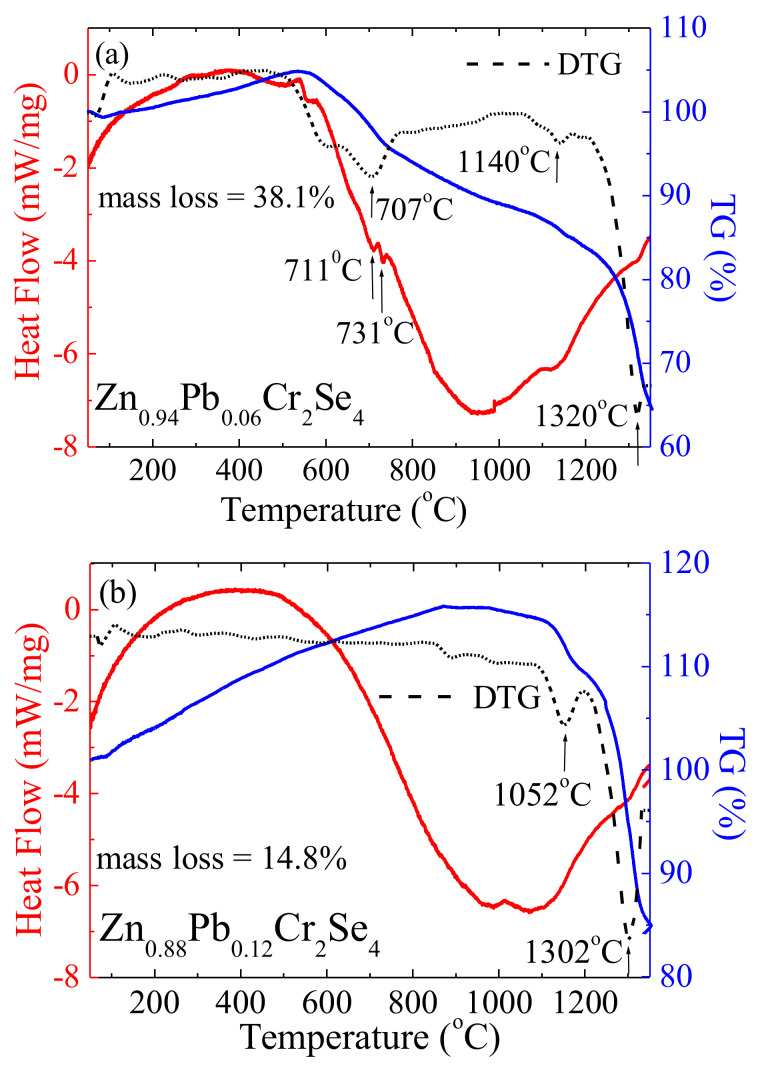
DSC/TG curves for Zn_0.94_Pb_0.06_Cr_2_Se_4_ (**a**) and Zn_0.88_Pb_0.12_Cr_2_Se_4_ (**b**) single crystals.

**Table 1 materials-15-05289-t001:** Conditions of growth and chemical composition of ZnCr_2_Se_4_:Pb single crystals.

N^o^	Amounts of Substrates (mmol)		*T*_d_ (K)	*T*_c_ (K)	Δ*T* (K)	% Weight				Chemical Formula
	n_ZnSe_	n_PbSe_				Zn	Pb	Cr	Se	
(1)	14.4	1.6	1223	1163	60	12.31 ± 0.09	3.23 ± 0.06	20.66 ± 0.01	63.71 ± 0.04	Zn_0.94_Pb_0.06_Cr_2_Se_4_
(2)	14.4	1.6	1133	1083	50	12.25 ± 0.08	2.95 ± 0.07	20.45 ± 0.02	64.54 ± 0.05	Zn_0.92_Pb_0.07_Cr_2_Se_4_
(3)	12.8	3.2	1203	1153	50	12.05 ± 0.06	3.43 ± 0.06	20.49 ± 0.05	64.10 ± 0.03	Zn_0.92_Pb_0.09_Cr_2_Se_4_
(4)	12.8	3.2	1203	1143	60	11.63 ± 0.02	4.51 ± 0.04	20.30 ± 0.06	63.56 ± 0.03	Zn_0.89_Pb_0.11_Cr_2_Se_4_
(5)	11.2	4.8	1203	1153	50	11.42 ± 0.03	4.98 ± 0.05	20.15 ± 0.03	63.42 ± 0.03	Zn_0.88_Pb_0.12_Cr_2_Se_4_

Chemical compositions were determined using Scanning Electron Microscopy. *T*_d_ is the temperature of the dissolution zone, *T*_c_ is the temperature of the crystallisation zone, and Δ*T* is the difference in temperatures between the dissolution and the crystallisation zones.

**Table 2 materials-15-05289-t002:** Structural parameters of the Zn_1-x_Pb_x_Cr_2_Se_4_ single crystals.

Chemical Formula	Lattice Parameter (Å)	Volume (Å^3^)	Density Calc. (Mg/m^3^)	Absorption Coeff. (mm^−1^)	Goodness of Fit on F^2^	R Parameters	
						R_1_	wR_2_
Zn_0.94_Pb_0.06_Cr_2_Se_4_	10.4910(3)	1154.65(2)	5.682	32.837	1.150	0.0223	0.0662
Zn_0.92_Pb_0.07_Cr_2_Se_4_	10.4933(2)	1155.41(3)	5.689	32.883	1.205	0.0333	0.1114
Zn_0.92_Pb_0.09_Cr_2_Se_4_	10.4946(4)	1155.84(1)	5.709	32.856	1.175	0.0261	0.0839
Zn_0.89_Pb_0.11_Cr_2_Se_4_	10.4954(2)	1156.10(2)	5.757	33.841	1.196	0.0333	0.1114
Zn_0.88_Pb_0.12_Cr_2_Se_4_	10.4959(1)	1156.27(3)	5.772	33.852	1.191	0.0311	0.1012

**Table 3 materials-15-05289-t003:** Atomic coordinates and equivalent isotropic displacement parameters of the Zn_1−x_Pb_x_Cr_2_Se_4_ single crystals. The Wyckoff positions of the atoms in the spinel structure are: Zn in 8 *b* (3/8, 3/8, 3/8); Cr/Ta in 16 *c* (½, ½, ½); and Se in 32 *e* (x, x, x). U (eq) is defined as one-third of the orthogonalized Uij tensor trace.

Spinel	Anion Parameter *u*	Site Occupation		U_iso_ (Å × 10^3^)		
		(A)	(B)	Zn/Pb	Cr	Se
(Zn_0.94_Pb_0.06)_[Cr_2_]Se_4_	0.2595(1)	0.94(1):0.06(1)	1.0	13(1)	7(1)	7(1)
(Zn_0.92_Pb_0.07_)[Cr_2_]Se_4_	0.2595(1)	0.92(2):0.07(1)	1.0	14(1)	6(1)	6(1)
(Zn_0.92_Pb_0.09_)[Cr_2_]Se_4_	0.2595(1)	0.92(1):0.09(1)	1.0	13(1)	7(1)	6(1)
(Zn_0.89_Pb_0.11_)[Cr_2_]Se_4_	0.2595(1)	0.89(2):0.11(2)	1.0	14(1)	5(1)	5(1)
(Zn_0.88_Pb_0.12_)[Cr_2_]Se_4_	0.2595(1)	0.87(1):0.13(2)	1.0	14(1)	5(1)	6(1)

**Table 4 materials-15-05289-t004:** Selected interatomic distances (Å) and bond angles (deg) of the Zn_1-x_Pb_x_Cr_2_Se_4_ spinel single crystals.

Spinel	Bond Distances		Bond Angles	
	Zn/Pb–Se	Cr–Se	Se–Zn/Pb–Se	Se–Cr–Se
(Zn_0.94_Pb_0.06_)[Cr_2_]Se_4_	2.4451(5)	2.5282(3)	109.5(0) × 6	180.0(0) × 3 94.61(13) × 6 85.39(13) × 6
(Zn_0.92_Pb_0.07_)[Cr_2_]Se_4_	2.4442(7)	2.5269(4)	109.5(0) × 6	180.0(0) × 3 94.61(2) × 6 85.39(2) × 6
(Zn_0.92_Pb_0.09_)[Cr_2_]Se_4_	2.4453(5)	2.5274(3)	109.5(0) × 6	180.0(0) × 3 94.62(14) × 6 85.38(14) × 6
(Zn_0.89_Pb_0.11_)[Cr_2_]Se_4_	2.4453(8)	2.5279(4)	109.5(0) × 6	180.0(0) × 3 94.62(2) × 6 82.38(2) × 6
(Zn_0.88_Pb_0.12_)[Cr_2_]Se_4_	2.4453(7)	2.5280(4)	109.5(0) × 6	180.0(0) × 3 94.61(2) × 6 85.39(2) × 6

**Table 5 materials-15-05289-t005:** Magnetic parameters of Zn_1−x_Pb_x_Cr_2_Se_4_ single crystals recorded in an internal oscillating magnetic field *H*_ac_ = 1 Oe with an internal frequency *f* = 120 Hz and with a zero external static magnetic field: *C* is the Curie constant; *T*_N_ and θ are the Néel and Curie–Weiss temperatures, respectively; µ_eff_ is the effective magnetic moment; *J*_1_ and *J*_2_ are the super-exchange integrals for the first two coordination spheres. *H*_c1_ and *H*_c2_ are the critical fields measured at the static magnetic field up to 70 kOe. *E*_a_ is the energy activation measured in the temperature range of 300–400 K. Experimental data for the ZnCr_2_Se_4_ matrix were taken from Refs. [1,21,22,31] for comparison.

Spinel	C	T_N_	θ	µ_eff_	M_s(2K)_	J_1_	J_2_	H_c1_	H_c2_	E_a_
	(emu·K/mol)	(K)	(K)	(µ_B_/f.u.)	(µ_B_/f.u.)	(K)	(K)	(kOe)	(kOe)	(eV)
ZnCr_2_Se_4_	4.082	21	90	5.714	6.0	−1.65	1.28	10.0	65.0	0.135
Zn_0.94_Pb_0.06_Cr_2_Se_4_	5.477	22	74	6.618	5.325	−2.07	1.17	12.0	55.0	0.138
Zn_0.92_Pb_0.07_Cr_2_Se_4_	5.568	22	77	6.673	5.964	−1.95	1.23	12.0	56.0	0.161
Zn_0.92_Pb_0.09_Cr_2_Se_4_	4.727	22	81	6.149	5.960	−2.02	1.19	12.0	61.0	0.146
Zn_0.89_Pb_0.11_Cr_2_Se_4_	5.768	22	83	6.792	5.486	−1.92	1.24	12.0	55.0	0.177
Zn_0.88_Pb_0.12_Cr_2_Se_4_	5.898	22	84	6.868	5.251	−1.90	1.25	12.0	57.0	0.315

**Table 6 materials-15-05289-t006:** Magnetic parameters of the Zn_1−x_Pb_x_Cr_2_Se_4_ single crystals recorded in an internal oscillating magnetic field *H*_ac_ = 1 Oe with an internal frequency *f* = 120 Hz and taken at external static magnetic fields *H*_dc_ = 0, 10, 20, 30, 40, and 50 kOe: *C* is the Curie constant; *T*_N_, θ, and *T*_m_ are the Néel, Curie–Weiss, and the spin fluctuation temperatures, respectively; µ_eff_ is the effective magnetic moment; and *J*_1_ and *J*_2_ are the super-exchange integrals for the first two coordination spheres.

Spinel	H_dc_	C	T_N_	θ	T_m_	µ_eff_	J_1_	J_2_
	(kOe)	(emu·K/mol)	(K)	(K)	(K)	(µ_B_/f.u.)	(K)	(K)
Zn_0.94_Pb_0.06_Cr_2_Se_4_	0	5.477	22	74	-	6.618	−2.07	1.17
	10	4.211	22	76	-	5.803	−1.73	1.13
	20	4.086	18	79	-	5.716	−1.38	1.11
	30	3.986	16	82	-	5.646	−1.03	1.08
	40	3.948	12	95	31.5	5.619	0.22	1.09
	50	3.863	6	108	46	5.558	0.90	1.05
Zn_0.92_Pb_0.07_Cr_2_Se_4_	0	5.568	22	77	-	6.673	−1.95	1.23
	10	4.182	20	78	-	5.783	−1.6	1.18
	20	4.102	19	81	-	5.728	−1.45	1.18
	30	4.003	16	84	-	5.658	−0.9	1.11
	40	3.988	12	96	33	5.648	−0.18	1.11
	50	3.976	7	99	44	5.639	0.60	1.00
Zn_0.92_Pb_0.09_Cr_2_Se_4_	0	4.727	22	81	-	6.149	−2.02	1.19
	10	3.922	21	82	-	5.601	−1.85	1.18
	20	3.711	20	84	-	5.448	−1.65	1.17
	30	3.681	16	85	-	5.426	−1.00	1.10
	40	3.602	12	97	33.5	5.367	−0.20	1.10
	50	3.639	6	108	46	5.369	0.90	1.05
Zn_0.89_Pb_0.11_Cr_2_Se_4_	0	5.768	22	83	-	6.792	−1.92	1.24
	10	4.293	20	84	-	5.859	−1.60	1.20
	20	4.182	19	85	-	5.783	−1.4	1.18
	30	4.101	16	86	-	5.727	−0.97	1.12
	40	4.056	12	91	34	5.695	−0.28	1.06
	50	4.002	6	97	48	5.657	0.72	0.96
Zn_0.88_Pb_0.12_Cr_2_Se_4_	0	5.898	22	84	-	6.868	−1.90	1.25
	10	3.794	20	86	-	5.508	−1.57	1.22
	20	3.781	19	88	-	5.499	−1.38	1.21
	30	3.762	16	90	-	5.485	−0.90	1.15
	40	3.703	12	94	36	5.442	−0.23	1.08
	50	3.689	6	98	48	5.432	0.73	0.97

## Data Availability

Not applicable.

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
