# Peer review of "The Zn1−xPbxCr2Se4—Single Crystals Obtained by Chemical Vapour Transport—Structure and Magnetic, Electrical, and Thermal Properties"

_materials, 2022, doi:10.3390/ma15155289_

Round 1

Reviewer 1 Report

Title: The Zn1-xPbxCr2Se4 - single crystals obtained by chemical vapour transport – structure and magnetic, electrical, and thermal properties

Authors: Izabela Jendrzejewska, Tadeusz Groń, Joachim Kusz, Zbigniew Stokłosa, Ewa Pietrasik, Tomasz Goryczka, Bogdan Sawicki, Jerzy Goraus, Josef Jampilek, Henryk Duda, Beata Witkowska-Kita

General Comments:

o  Single crystals with the general formula Zn1-xPbxCr2Se4 (x = 0.06, 0.07, 0.09. 0.11 and 0.12), where Pb2+- ions are located in the tetrahedral sites of the spinel structure, were successfully synthesized and their structural, electric and magnetic properties were analyzed.

o   The structure of the article fulfills the structure of a research article.

o   Five keywords are included by the authors.

  • The Introduction section should be rewritten because it contains almost only self-citations (23 references from 34 are self-citations, respectively 67.64% of references are self-citation).

o   The authors present in the Materials and Methods section shortly the reagents, the experimental method and the equipment used.

o   In the Results section, the authors present and interpret the results of the performed experiments.

o   The paper ends with the Conclusions part. In this section the authors mention the conclusions of their research study.

I suggest to Reconsider after Major Revisions for the following reasons:

1. 67.64% of references are self-citation: I can understand that the authors have a vast expertise in the field, however the number of self-citations is huge;

2. the chemical elements are written with both symbols and the full name - please check the whole document and keep only one way of writing;

3. page 3, line 108-109: please correct the chemical reaction equations;

4. page 3, line 140: ref 25 is cited twice;

5. page 6, line 183-184: please rephrase the sentence;

6. page 7: please provide the diffraction pattern;

7. page 15, line 382: please replace “has appeared” with “start at”….

8. page 15, line 387: delete “especially” – the mass loss can be observed only on TG curve.

Author Response

A detailed response to the Reviewers’ comments (Materials-1781147)

We thank the Reviewers for their careful reading of the paper and constructive remarks. In order to take into account the latter, the paper has been revised. The spelling has been checked. All changes are marked in yellow.

Reviewer 1:

General Comments:

â—‹ The Introduction section should be rewritten because it contains almost only self-citations (23 references from 34 are self-citations, respectively 67.64% of references are self-citation).

A: The Introduction section has been improved, 16 external citations have been added. Now self-citations are only for 46% (23 out of 50). The number of publications describing spinel materials based on the ZnCr2Se4 matrix is enormous. The choice is not easy.

Major Revisions for the following reasons:

  1. 64% of references are self-citation: I can understand that the authors have a vast expertise in the field, however the number of self-citations is huge;

A: The ratio has been improved and is now 46%. In the work of D.R. Parker et al. published in J. Am. Chem. Soc. 126 (2004) 2710-2711 the authors wrote that: “The pioneering work of the Katowice groups [9-15] and that of other groups [16-21] has, over many years, revealed a rich and complex behaviour in this series”.

  1. the chemical elements are written with both symbols and the full name - please check the whole document and keep only one way of writing;

A: Chemical elements are written with symbols throughout the whole document.

  1. page 3, line 108-109: please correct the chemical reaction equations;

A: This chemical reaction has been corrected

  1. page 3, line 140: ref 25 is cited twice;

A: Double citation Ref. 25 has been removed.

  1. page 6, line 183-184: please rephrase the sentence;

A: The sentence has been reformulated: "Based on these calculations, the reaction conditions, such as dissolution and crystallization temperatures, and their difference were determined."

  1. page 7: please provide the diffraction pattern;

A: To show the diffraction pattern of our crystals, we destroyed one crystal – the crystal was powdered in agate mortar and then examined using X’Pert Philips diffractometer (PW 1050). In the picture, we present the diffraction pattern of Zn0.89Pb0.11Cr2Se4-single crystals with the diffraction pattern of pure ZnCr2Se4 for comparison. 

It is worth highlighting that the crystals obtained in chemical vapour transport grow in the zone where the reagents are not present (scheme below). Such crystal, which possesses well-formed walls with edges, is a single-phase compound without other phases.

This scheme was presented in our previous work [41].

  1. page 15, line 382: please replace “has appeared” with “start at”….

A: The words have been replaced.

  1. page 15, line 387: delete “especially” – the mass loss can be observed only on TG curve.

A: The word "especially" has been deleted.

Reviewer 2 Report

In this manuscript, Jendrzejewska et al present an experimental study of Pb-doped ZnCr2Se4.  ZnCr2Se4 is a semiconducting antiferromagnetic spinel with potential applications in the field of spintronics.  The authors have synthesized single crystal samples of Zn(1-x)Pb(x)Cr2Se4 (x = 0.06 to 0.12), and characterized the structural, magnetic, electrical and thermal properties of these samples.  They employ a broad range of experimental techniques, including x-ray diffraction, scanning electron microscopy, differential scanning calorimetry, thermogravimetric analysis, SQUID magnetometry, and electrical conductivity, and specific heat.  The main conclusions from this article are:

- Single crystal samples of Pb-doped ZnCr2Se4 can be synthesized using Chemical Vapour Transport methods.

- Like the parent compound, these compounds are semiconducting antiferromagnets. 

- In the presence of applied magnetic field, antiferromagnetic interactions appear to be suppressed, and there is an increasing tendency towards ferromagnetism.

The authors present an impressive amount of experimental characterization data in this paper.  Pb-doped ZnCr2Se4 appears to be a new doped system, and this paper helps to map out the structural, magnetic, electrical, and thermal effects of doping.

However, the motivation for studying Pb-doping is not very strongly presented.  The focus of this manuscript is quite narrow (i.e. it almost exclusively consists of describing or comparing basic characterization measurements of doped ZnCr2Se4), and the authors heavily engage in self-citation (23 out of 34 references involve co-authors from the current manuscript).  Given the extent of their previous work on this family of materials, some self-citation is certainly justified, but the current reference list does not appear to demonstrate a broad interest in these compounds. 

Although the Pb-doped compound is a new addition to this series of doped compounds, it it not clear how much this study adds to our understanding of ZnCr2Se4.  It would be helpful if the authors could emphasize why this doping series is interesting, or how it can improve our knowledge of ZnCr2Se4, spinels, or frustrated materials in general.

Additional comments and concerns:

1. I would recommend that the abstract be heavily revised/rewritten.  At present, it consists of a long list of characterization results, largely presented in a single run-on sentence.  This would be a good opportunity to motivate the current study, and highlight the most important findings.

2. I would recommend that the authors provide a more detailed description of experimental methods.  In particular, the authors include no details of the SEM measurements, and the only description of the x-ray diffraction measurements and the determination of chemical composition is a reference to two previous papers by the authors of this study.  Given the small differences between doping levels in this study (x = 0.06, 0.07, 0.09, 0.11, and 0.12), it is very important to demonstrate that the stoichiometry of these samples has been accurately determined.

3. On Line 140 there are multiple references to Ref. 25 in the same citation.

4. On Line 165 there is a division symbol which should likely be a hyphen.

5. On Lines 183-184 there are missing words (or a sentence fragment).

6. In the caption to Table I it should be stated how the chemical compositions have been determined.

7. In Section 3.2 the authors describe two structural models: allowing Pb dopants to occupy either the Zn or Cr sites.  Did they explore any models which allow both of these types of substitution?

8. Did the authors attempt to grow any samples with higher concentrations of Pb?  Is there a reason no samples were synthesized beyond x = 0.12?

9. In Fig. 4 the relative uncertainties in the values of Pb concentration (x) are much larger than the relative uncertainties in the lattice parameters.  Therefore, it would be helpful to include horizontal error bars in this plot as well.

10. On Lines 233-234 the authors refer to the Se positional parameter as both u and x.  This makes the text a bit confusing, therefore I would recommend picking one symbol and sticking with it.

11. It is strange that the magnetic and electrical parameters (e.g. Tn, mu, theta, and Ea) do not vary more smoothly or systematically as a function of doping.  Do the authors have a good explanation for why this is the case?  This might suggest that there are some larger uncertainties in sample stoichiometry or inhomogeneities in doping concentration.

Author Response

A detailed response to the Reviewers’ comments (Materials-1781147)

We thank the Reviewers for their careful reading of the paper and for their constructive remarks. In order to take into account the latter, the paper has been revised. The spelling has been checked. All changes are marked in yellow.

Reviewer 2:

However, the motivation for studying Pb-doping is not very strongly presented.  The focus of this manuscript is quite narrow (i.e. it almost exclusively consists of describing or comparing basic characterization measurements of doped ZnCr2Se4), and the authors heavily engage in self-citation (23 out of 34 references involve co-authors from the current manuscript).  Given the extent of their previous work on this family of materials, some self-citation is certainly justified, but the current reference list does not appear to demonstrate a broad interest in these compounds.

A: Pb2+ is interesting for research because it is a diamagnetic ion with a large ion radius compared to zinc. The number of self-citations decreased to 46% (23 out of 50) after adding the 16 external citations in the Introduction. The number of publications describing spinel materials based on ZnCr2Se4 matrix is enormous, the choice is not easy. However, our spinel research efforts have been noticed since in the work of D.R. Parker et al. published in J. Am. Chem. Soc. 126 (2004) 2710-2711 the authors wrote that: “The pioneering work of the Katowice groups [9-15] and that of other groups [16-21] has, over many years, revealed a rich and complex behavior in this series”.

Although the Pb-doped compound is a new addition to this series of doped compounds, it it not clear how much this study adds to our understanding of ZnCr2Se4.  It would be helpful if the authors could emphasize why this doping series is interesting, or how it can improve our knowledge of ZnCr2Se4, spinels, or frustrated materials in general.

A: This work shows how the appropriate doping and measurement methods and the applied ac and dc magnetic fields emphasize the type of magnetic interactions and their range.

Additional comments and concerns:

  1. I would recommend that the abstract be heavily revised/rewritten. At present, it consists of a long list of characterization results, largely presented in a single run-on sentence.  This would be a good opportunity to motivate the current study, and highlight the most important findings.

A: The abstract has been changed and completely redrafted.

  1. I would recommend that the authors provide a more detailed description of experimental methods. In particular, the authors include no details of the SEM measurements, and the only description of the x-ray diffraction measurements and the determination of chemical composition is a reference to two previous papers by the authors of this study. Given the small differences between doping levels in this study (x = 0.06, 0.07, 0.09, 0.11, and 0.12), it is very important to demonstrate that the stoichiometry of these samples has been accurately determined.

A: The technical details of SEM and X-ray study were added to the main text. The chemical composition of the crystal is calculated based on values of  %weight for each element that forms the crystal. The basis for the calculations is the assumption that there are 4 Se-ions, because Se-ions form an F-type unit cell, and in the tetrahedral and octahedral voids, there are Zn and Cr ions, respectively. Then, the chemical composition is refined using an X-ray diffraction study.

  1. On Line 140 there are multiple references to Ref. 25 in the same citation.

A: Double citation Ref. 25 has been removed.

  1. On Line 165 there is a division symbol which should likely be a hyphen.

A: The division symbol has been replaced with a hyphen.

  1. On Lines 183-184 there are missing words (or a sentence fragment).

A: The sentence has been reformulated and reads as follows: "Based on these calculations, the reaction conditions, such as dissolution and crystallization temperatures, and their difference were determined."

  1. In the caption to Table I it should be stated how the chemical compositions have been determined.

A: In the caption to Table 1, a sentence marked with an asterisk has been added: “Chemical compositions were determined using Scanning Electron Microscopy.”

  1. In Section 3.2 the authors describe two structural models: allowing Pb dopants to occupy either the Zn or Cr sites. Did they explore any models which allow both of these types of substitution?

A: The study of models in which Pb is incorporated in position B, i.e. in place of Cr, was abandoned for two reasons: the refinement of the placement of Cr gave the value of 1.0 within the error limits and because the Pb atom, being in this position, would have to be trivalent.

  1. Did the authors attempt to grow any samples with higher concentrations of Pb?  Is there a reason no samples were synthesized beyond x = 0.12?

A: We tried to obtain the single crystals with larger Pb, but our attempts were unsuccessful. It is several factors, which influence the amount and cation distribution, for example: ionic radius, polarisation, crystal lattice energy, and crystal fields theory. In our case, it is very likely, that the main reason is a structural factor, i.e. the large Pb2+- ion radius.

  1. In Fig. 4 the relative uncertainties in the values of Pb concentration (x) are much larger than the relative uncertainties in the lattice parameters.  Therefore, it would be helpful to include horizontal error bars in this plot as well.

A: This figure has been corrected.

  1. On Lines 233-234 the authors refer to the Se positional parameter as both u and x.  This makes the text a bit confusing, therefore I would recommend picking one symbol and sticking with it.

A: This mistake has been corrected.

  1. It is strange that the magnetic and electrical parameters (e.g. Tn, mu, theta, and Ea) do not vary more smoothly or systematically as a function of doping.  Do the authors have a good explanation for why this is the case?  This might suggest that there are some larger uncertainties in sample stoichiometry or inhomogeneities in doping concentration.

A: Magnetic parameters such as the Néel temperature and the paramagnetic Curie-Weiss temperature are generally not sensitive to the change in Pb content in the sample, because the only magnetic ion is the Cr3+ ion, the concentration of which does not change. The situation is similar in the case of electric conductivity, because Pb2+ ions occupying tetrahedral sites in the spinel structure additionally have 4f, 5d and 6s subshells filled. The ZnCr2Se4 matrix itself is a semiconductor with a predominance of ionic bonding. The slight differences in electrical conductivity result from structural defects and vacancies since it is known that in a state of thermal equilibrium, structural defects are always present in the lattice, even in the crystal, which is ideal in other respects.

Reviewer 3 Report

1. two figure 4 detected. correct it. 

2. Please show the as grown crystal image in addition to what you have shown.

3. How do you ensure that the presence of lead in the crystal lattice? since it is high temperature sintering method explain the possibility of getting dopant in the lattice

Author Response

A detailed response to the Reviewers’ comments (Materials-1781147)

We thank the Reviewers for their careful reading of the paper and for their constructive remarks. In order to take into account the latter, the paper has been revised. The spelling has been checked. All changes are marked in yellow.

Reviewer 3:

Comments and Suggestions for Authors

  1. two figure 4 detected. correct it.

A: The numbering of figures has been corrected.

  1. Please show the as grown crystal image in addition to what you have shown.

A: We added the photo of quartz ampoule after synthesis (Fig. 4a), in which we see ZnCr2Se4:Pb single crystals. The synthesis takes place in closed system, inside the horizontal pipe furnace. We cannot observe the process of growth. We only can see the crystals after synthesis.

  1. How do you ensure that the presence of lead in the crystal lattice? since it is high temperature sintering method explain the possibility of getting dopant in the lattice

A: We investigate our single crystals using X-ray methods: SEM and diffraction. Based on these study, the chemical composition and cation distribution is determined. It is worth highlighting that the methods of single crystal growth, called “chemical vapour transport, " are not the same as the high-temperature sintering of powders. We use the high-temperature sintering of powders to obtain the binary selenides, such as ZnSe and PbSe, etc.

        The chemical vapour transport is the transfer of a solid via the gas phase from an area with a temperature T1 to an area with a different temperature T2 using a transporting substance. This substance will fulfil its task only when a heterogeneous reversible reaction occurs between it and the solid, resulting in only gaseous products forming.

There are three main steps in the chemical transport process:

  1. transfer of substance A(s) into the gas phase at a temperature T1 (dissolution zone);
  2. moving the gaseous reaction products from the T1 area to the T2 area; this mixture is transferred to the T2 area by diffusion, convection or forced gas flow.
  3. crystallization of substance A(s) at T2; at T2, the crystallization of substance A(s) takes place, resulting in the saturated solution becoming unsaturated. The gaseous solution returns to the dissolution zone, replenishes the loss of substance A (s), and again goes as a supersaturated solution to the crystallization zone.

         Below, we present the scheme of chemical vapour transport. This scheme was shown in our previous papers [41], for this reason, we did not give this picture in this work.

Round 2

Reviewer 1 Report

The manuscript has been sufficiently improved to warrant publication in Materials.